# AdS black holes with a bouncing interior

Sean A. Hartnoll[1] and Navonil Neogi[2]

**1** Department of Applied Mathematics and Theoretical Physics, University of Cambridge, Cambridge CB3 0WA, United Kingdom
**2** Trinity College, Cambridge CB2 1TQ, United Kingdom

## Abstract

We construct planar black hole solutions of AdS gravity minimally coupled to a scalar field with an even, super-exponential potential. We show that the evolution of the black hole interior exhibits an infinite sequence of Kasner epochs, as the scalar field rolls back and forth in its potential. We obtain an analytic expression for the 'bounces' between each Kasner epoch and also give an explicit formula for the times and strengths of the bounces at late interior times, thereby fully characterizing the interior evolution. In this way we show that the interior geometry approaches the Schwarzschild singularity at late times, even as the scalar field is driven higher up its potential with each bounce.



## 1 Introduction

The role of black hole interiors in Anti-de Sitter (AdS) holography remains to be fully elucidated. The interior is causally disconnected from the asymptotic boundary and yet is determined by initial data in the exterior. Previous works have related black hole interiors to

important dynamical processes in the dual quantum many-body system [1–5]. However, the rich classical dynamics of black hole interiors — such as the emergence of BKL chaos [6–8] — has yet to find a holographic home. Conversely, the dual field-theoretic description holds out the prospect of understanding stringy and quantum gravitational effects at the interior singularity, but these have proven challenging to nail down [9, 10].

A widely employed simplification in the study of cosmological dynamics, and in particular the approach to singularities, is minisuperspace. This amounts to an ansatz in which the fields depend on time but not space. It has recently been emphasized that interior minisuperspace dynamics is naturally thought of as the continuation of the exterior holographic renormalization group flow through the horizon [11, 12]. While solutions obtained by neglecting spatial inhomogeneities in the interior will be highly non-generic, at late times spatial gradient terms are expected to drop out of the equations of motion [6–8]. This implies that different points in space decouple and each separately evolve according to the minisuperspace equations.

A recent body of work has demonstrated that a common feature of (minisuperspace) holographic interiors is the emergence of Kasner scaling towards the singularity [11, 13–29]. The negative cosmological constant becomes irrelevant at late interior times, and therefore the well-studied approach to spacelike singularities in gravity without a cosmological constant is recovered. A further phenomenon that appears in these works are bounces between different Kasner 'epochs', see e.g. [14, 16, 21, 25, 27–29]. This is again familiar from much earlier work, most famously the chaotic bounces of the 'mixmaster' universe [30].

In this paper we obtain a holographic model of a black hole with an infinitely bouncing interior, with the key aspects of the dynamics captured analytically. As part of this endeavour, we characterize the cosmological behavior of gravity with super-exponential potentials — which has some differences to the well-studied case of exponential potentials [7, 30].

Let us briefly outline the main technical results. We will find planar black hole solutions in which the metric and a scalar field evolve from the AdS boundary to the black hole interior as a function of a single 'radial' coordinate, in the spirit of [11]. The metric ansatz is given in equation (2). The evolution can be reduced to solving a single second-order, nonlinear ordinary differential equation for the velocity of the scalar field — equation (12). If the scalar field has an even, super-exponential potential then it cannot reach an asymptotic scaling behavior [16] and necessarily bounces back and forth in its potential; this evolution of Kasner epochs punctuated by 'bounces' is illustrated in Fig. 1. At the bounces the scalar field runs high up its potential and the full equation of motion can be simplified to equation (20). As previously noted in [28], this equation provides the analytic expression (24) for the bounces between epochs. Using the simplified equation for the bounces we demonstrate that the velocity of the scalar field relaxes at each bounce, while the field itself is driven higher and higher up its potential. By obtaining the exponentially small variation of the velocity in the Kasner regimes, in between bounces, we are able to set up recursion relations in (32) that characterize the evolution of the field and its velocity from bounce to bounce. Solving these recursion relations, we show that at late interior times the velocity of the field tends to zero and the interior metric tends towards the Schwarzschild singularity. All analytical expressions are corroborated by numerical solution to the equations of motion.

## 2 Equations

We consider gravity minimally coupled to a scalar field in $3+1$ bulk dimensions. We allow the scalar field to have an arbitrary potential, so that the Lagrangian density is

$$\mathcal{L} = R - \tfrac{1}{2} g^{ab} \partial_a \phi \partial_b \phi - V(\phi). \tag{1}$$

We have set the gravitational coupling to one. The negative cosmological constant is included within the definition of $V(\phi)$. We wish to study planar black hole solutions to the theory (1) that have the form

$$ds^2 = \frac{1}{z^2}\left(-f(z)e^{-\chi(z)}dt^2 + \frac{dz^2}{f(z)} + dx^2 + dy^2\right), \qquad \phi = \phi(z). \tag{2}$$

The AdS boundary is at $z = 0$ and the singularity will be at $z \to \infty$. At a horizon, $f = 0$. We will take the potential to have the asymptotic behavior as $\phi \to 0$ near the AdS boundary

$$V(\phi) \to -6 - \phi^2 + \cdots. \tag{3}$$

This fixes the asymptotic AdS radius to unity and the scaling dimension of the scalar field to $\Delta = 2$, using standard quantization. It follows that to leading asymptotic behavior at the AdS boundary as $z \to 0$:

$$f \to 1, \qquad \chi \to 0, \qquad \phi \to \phi_{(0)} z. \tag{4}$$

Here $\phi_{(0)}$ is the boundary source for the scalar field. The asymptotic value of $\chi$ may be set to zero by rescaling the time coordinate. The well-established holographic dictionary, e.g. [31], relates the setup just described to a three dimensional conformal field theory at nonzero temperature and deformed by a relevant operator $\mathcal{O}$, dual to the bulk field $\phi$.

It is straightforward to verify that the Einstein and scalar field equations are solved on the ansatz (2) so long as

$$2z\chi' = z^2\phi'^2, \tag{5}$$

$$4zf' = \left(12 + z^2\phi'^2\right)f + 2V(\phi), \tag{6}$$

$$2z^2f\phi'' = z\phi'\left(-2zf' + [4 + z\chi']f\right) + 2V'(\phi). \tag{7}$$

These equations imply a single third order, nonlinear equation for $\phi$. Before writing down this equation it will be helpful to parametrize the potential as

$$V(\phi) = e^{\int H(\phi)d\phi}, \tag{8}$$

for some function $H(\phi)$ and introduce the coordinate $\rho$ via

$$z = e^\rho. \tag{9}$$

At the AdS boundary $\rho \to -\infty$ while at the singularity $\rho \to +\infty$. The equation for $\phi$ is then obtained from (5) – (7) as

$$4[2H(\phi) - \dot\phi]\frac{\dddot\phi}{\dot\phi} = 8[H(\phi)]^2\dot\phi - 8\ddot\phi + 4[3 + 2H'(\phi)]\dot\phi + \dot\phi^3 - 6H(\phi)(4 + \dot\phi^2). \tag{10}$$

Here dots denote derivatives with respect to $\rho$. Finally, this equation may be written as a second order equation for $v(\phi)$ where the 'velocity'

$$v \equiv \dot\phi. \tag{11}$$

Equation (10) then becomes

$$4(2H - v)\frac{vv''}{v'} = 8H^2v - 2H[3v^2 + 4(3 + v')] + v[12 + v^2 + 4(2H - v)']. \tag{12}$$

In this equation both $v$ and $H$ are functions of $\phi$. Our objective is to understand the behavior of (10) or (12) at late interior times.

## 3 Kasner epochs and numerics

In the absence of a potential $V$ it has been long known [32] that at late interior times $\rho \to +\infty$ the solution will evolve towards a Kasner spacetime with the velocity

$$v = v_\infty \,, \tag{13}$$

a constant. On this solution the field $\phi = v_\infty \rho \to +\infty$ at late interior times if $v_\infty > 0$. The other fields are then, from the equations of motion above and ignoring the potential,

$$\dot{\chi} = \frac{v_\infty^2}{2} \,, \qquad \frac{\dot{f}}{f} = 3 + \frac{v_\infty^2}{4} \,. \tag{14}$$

In particular, $v_\infty = 0$ describes the Schwarzschild singularity, with no scalar field.

Kasner spacetimes continue to play a central role once a potential is included, but they may not persist for ever. To see this, consider a perturbation of the Kasner behaviour

$$v = v_\infty + \delta v(\phi) \,. \tag{15}$$

Linearising equation (12) in $\delta v$ and integrating gives

$$\delta v(\phi) = a \int e^{-(3+v_\infty^2/4)\phi/v_\infty} \left[ V(\phi) - \frac{2}{v_\infty} V'(\phi) \right] \mathrm{d}\phi \,. \tag{16}$$

Here $a$ is a constant. Suppose firstly that $V(\phi)$ is sub-exponential as $\phi \to \infty$. In this case the integral in (16) is dominated at large $\phi$ by the first, exponentially decaying term. The perturbation $\delta v(\phi)$ therefore remains small and the Kasner asymptotic survives. However, if the potential is super-exponential then the $V(\phi)$ terms dominate the integral in (16) at large $\phi$, and therefore $\delta v(\phi)$ becomes large. It follows that in such cases Kasner is not a stable asymptotic behaviour for the interior [16]. When the potential is precisely exponential, then it is possible for asymptotic Kasner behavior to survive by self-consistently approaching a value of $v_\infty$ such that the negative exponential term in (16) dominates over the potential.

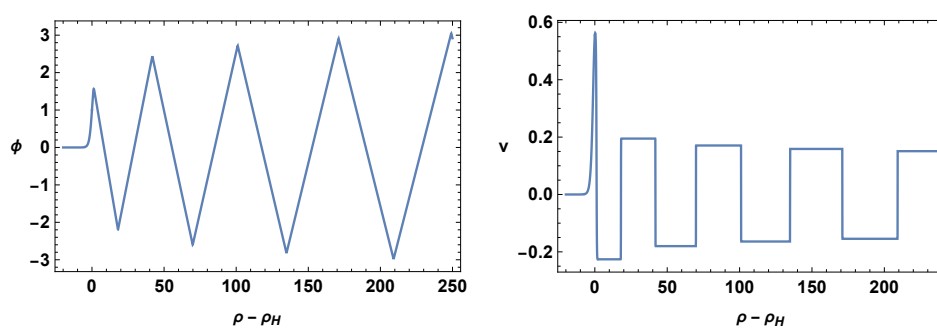

Figure 1: Evolution of $\phi$ (left plot) and $v = \dot{\phi}$ (right plot) as a function $\rho$. The AdS boundary is at $\rho \to -\infty$, the horizon at $\rho = \rho_H$ and the singularity at $\rho \to +\infty$. Numerics have been performed with the potential $V = -6 - \phi^2 + \frac{1}{10} \exp\left[\frac{1}{10}\phi^8\right]$. Towards the boundary $\phi \sim \dot{\phi} \sim e^\rho$. The constants of integration have been fixed by regularity at the horizon combined with $\phi(\rho_H) = 1$. Numerical evolution with a super-exponential potential is delicate. We have found it most convenient to integrate the original equations (5) – (7), converted to the $\rho$ coordinate. See e.g. [11] and references therein for discussion of numerical methods.

In the remainder of this paper we will determine, analytically, the asymptotic behavior of the interior in cases where $V(\phi)$ is super-exponential at large $\phi$. We restrict to even potentials so that the field may not escape to infinity in either direction. We may first, however, gain some intuition from numerics. Fig. 1 shows an illustrative evolution of the scalar field $\phi$ and its derivative $v = \dot{\phi}$ from the boundary to the interior. Similar bouncing behaviour has previously been seen numerically in [16]. We have chosen a strong potential that rapidly develops several Kasner epochs in the interior.

The interior dynamics in Fig. 1 exhibits a sequence of Kasner epochs characterized by decreasing constant velocities $v_1 > v_2 > v_3 > \cdots$ The epochs are separated by abrupt 'bounces' in which the velocity changes sign. The maximal value of the scalar field increases from bounce to bounce. The relatively small, and decreasing, values of the Kasner velocities suggests that the solution is slowly approaching the Schwarzschild singularity at late interior times. This fact is furthermore illustrated in Fig. 2, and will be shown to be the case in §5.

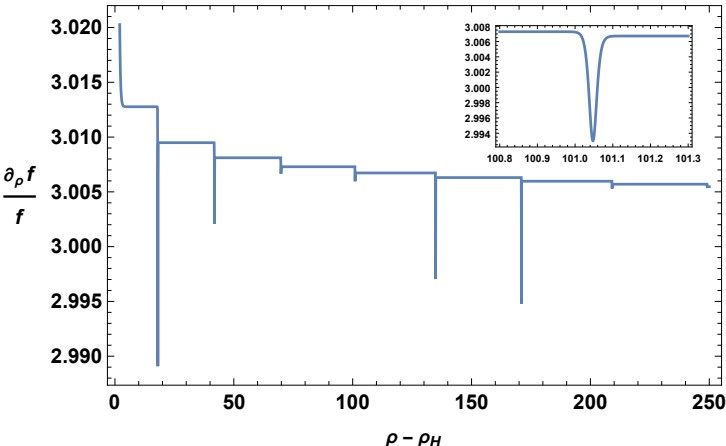

Figure 2: The metric slowly approaches the Schwarzschild singularity behaviour $\partial_\rho f / f = 3$ at late interior times, through a sequence of Kasner epochs. The sharp dips in $\partial_\rho f / f$ at the transitions between epochs are not spurious. The inset zooms in on one of the transitions.

## 4 Analytic description of the extended bounce

While the scalar field and its derivative never become large, there are large numbers appearing in the equations. Close to the transitions the potential $V$ and its 'exponent' $H$, evaluated on the solution $\phi(\rho)$, become large. This is illustrated in Fig. 3, using the numerics from the previous section. Recall that it is only $H$ and its derivatives that appear in the scalar equations of motion (10) or (12). It is important to note in Fig. 3 that while $V$ is large very close to the transition, and this will be the narrow region where the velocity $v$ changes sign, $H$ remains (less) large over a wider range of values of $\rho$. We will call this the 'extended bounce' region.

An analytic understanding of the bounces between Kasner epochs is obtained by identifying the terms in the equation of motion (12) that dominate throughout the 'extended bounce' regimes. Figs. 1 and 3 together suggest that these regimes have

$$|H| \gg |v|. \tag{17}$$

The numerics furthermore indicate, see Fig. 4 below, that this condition continues to hold well into the Kasner epochs on each side of the transition, where the velocity $v$ has become

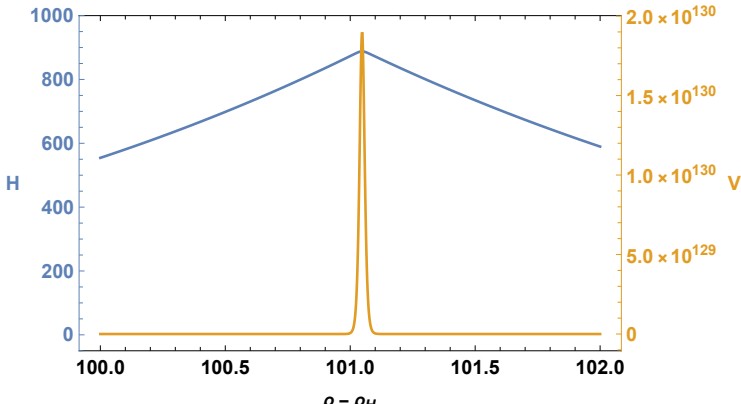

Figure 3: The potential $V(\phi(\rho))$ (right axis, orange) and $H(\phi(\rho)) = V'/V$ (left axis, blue) close to one of the transitions between Kasner epochs shown in Fig. 1. The potential varies much more abruptly than $H$. This leads to a narrow as well as an extended bounce regime.

constant. The overlap of the extended bounce with the constant $v$ epochs will be important later, as it will allow us to match the Kasner epochs onto the far narrower regime over which $v$ changes sign.

With the assumption (17) the equation of motion (12) simplifies to

$$\frac{v''}{v'} + \frac{v'}{v} = H - \frac{3}{v} + \frac{H'}{H}. \tag{18}$$

We see that this simplified equation indeed allows for regimes where velocity terms dominate — these will be the onset of the Kasner epochs where the terms involving $H$ become negligible — as well as regimes where the potential dominates. From (8) we have that $H = V'/V$ and hence we may integrate (18) once to obtain

$$\log \frac{vv'}{V'} = -3 \int \frac{\mathrm{d}\phi}{v} = -3\rho + \text{const}. \tag{19}$$

We used the definition of $v$ from (11) in the final step. Also from (11) we have that $vv' = \ddot{\phi}$ and therefore we may write

$$\ddot{\phi} = cV'(\phi)e^{-3\rho}. \tag{20}$$

Here $c$ is a constant. In principle this constant could be different for each bounce, as the condition (17) does not hold all the way from one bounce to the next. In particular, $H$ changes sign and therefore vanishes in between bounces. The matching argument in §5 below, however, will show that $c$ is in fact the same on each bounce.

Equation (20) may also, of course, be obtained by integrating up (10) subject to the conditions (17). We have found the above derivation cleaner.

The greatly simplified equation of motion (20) describes a particle moving in a potential $V(\phi)$ that is uniformly shrinking as a function of time due to the factor of $e^{-3\rho}$. This is very similar to the situation in which bounces between Kasner epochs were first described in the context of the mixmaster universe [30]. Let us first recall qualitatively how this works. On the $n$th Kasner epoch the scalar field grows according to $\phi = v_n\rho$, and therefore the right hand side in (20) is $V'(v_n\rho)e^{-3\rho}$. Because $V$ is super-exponential, eventually the growth in $V'$ dominates the exponential shrinking. This is entirely analogous to our discussion around (16) above. Once the potential term dominates, it is clear what will happen: the scalar field will bounce off the potential. As the field starts to roll back down the potential, the shrinking factor

of $e^{-3\rho}$ will rapidly come to dominate. The field will then lose sight of the potential and will again acquire a 'free motion' Kasner behavior $\phi = v_{n+1}\rho$. Because we are taking the potential to be even, this new Kasner regime is again unstable and the process will repeat itself for ever, consistent with the numerics in §3.

To obtain the form of the bounce analytically, one further approximation is necessary. From (20) it is immediate to obtain

$$\frac{\dddot{\phi}}{\ddot{\phi}} + 3 = \frac{V''(\phi)}{V'(\phi)}\dot{\phi}\,. \tag{21}$$

Eq. (21) depends only on the rate of growth of the potential. From Fig. 3 we know that this rate of growth varies slowly compared to the abruptness of the bounce itself. Thus the rate of growth is approximately constant over the bounce, so that at the $n$th bounce $V''/V' \to k_n$[1] and we may simplify (21) to

$$\frac{\dddot{\phi}}{\ddot{\phi}} + 3 = k_n\dot{\phi}\,. \tag{23}$$

Eq. (23) amounts to approximating the potential close to the bounce by the exponential form $V' \propto e^{k_n\phi}$, which is sufficient to capture the competition in (20) between the growth of V with $\phi$ and the shrinking of the effective potential with $\rho$. Eq. (23) was previously found to describe bounces off exponential potentials in [28], who also noted its solution

$$\dot{\phi} = \frac{3}{k_n} - \frac{|\Delta v_n|}{2}\tanh\left[\frac{k_n|\Delta v_n|}{4}(\rho - \rho_n)\right]\,. \tag{24}$$

There are two constants of integration in this solution. The change in the Kasner velocity at the $n$th bounce is $v_{n+1} - v_n = -|\Delta v_n|\operatorname{sgn}(k_n)$, which occurs at $\rho = \rho_n$ (this is where $\dddot{\phi} = 0$, we may alternatively think of the bounce as occurring where $\ddot{\phi} = 0$. Because $k_n$ is large in (24) these two points are close together). In Fig. 4 we verify that the expression (24) gives an excellent description of the bounces found numerically in §3. It may be noted that the equation (23) and solution (24) are somewhat similar, but not quite identical, to the approximate analytic description of 'Kasner inversions' of charged black holes obtained in [14,33]. A difference with the Kasner inversions is that here the metric component $g_{tt}$ continues to collapse towards zero across all transitions.

In Fig. 4 we see that the solution (24) describes a bounce between two Kasner epochs

$$\dot{\phi} = v_n\,, \quad \text{for} \quad \rho \ll \rho_n\,, \qquad \to \qquad \dot{\phi} = v_{n+1} \quad \text{for} \quad \rho_n \ll \rho\,. \tag{25}$$

One observation that follows from (24) is that while the Kasner velocities may often change sign, the magnitude always decreases from bounce to bounce: $|v_n| \geq |v_{n+1}|$, consistently with Fig. 1 above. We may think of the reduction in velocity as some 'inelasticity' in the bounce although, possibly counterintuitively, we will also establish shortly that the field $\phi$ rolls further up the potential on each subsequent bounce. This phenomenon is also visible in Fig. 1. In the following section we will show that $|v_n| \to 0$ as $n \to \infty$, so that the solution eventually relaxes to the Schwarzschild singularity.

---

[1]More precisely, the condition to make this replacement is that $|\Delta(V''/V')| \ll |V''/V'|$ over the range $\Delta\phi$ of the narrow bounce. The range can be estimated from (21) as $\Delta\phi \sim V'/V''$. Using $\Delta V'' \sim V'''\Delta\phi$ and $\Delta V' \sim V''\Delta\phi$ the required condition is that

$$\left|\frac{V'''V'}{V''^2} - 1\right| \ll 1\,. \tag{22}$$

This condition is seen to hold at large $\phi$ for the kind of potentials we are considering.

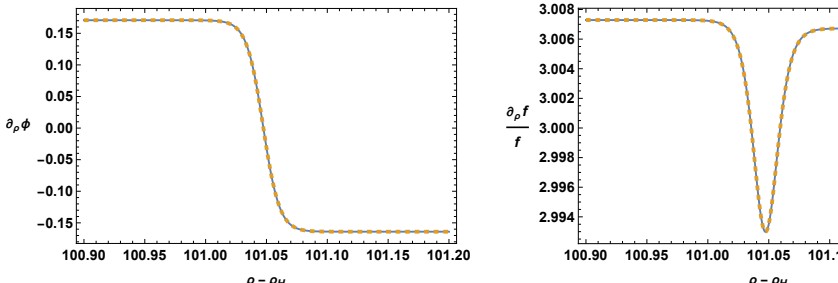

Figure 4: Evolution of $\dot{\phi}$ (left) and $\dot{f}/f$ (right) across a bounce. Orange dashes are the full numerical solution (cf. inset of Fig. 2 for $\dot{f}/f$). Solid blue lines show the analytic expressions following from a fit to (24). This is a two-parameter fit for $\rho_n$ and $|\Delta v_n|$, with $k_n = V''/V'$ at $\rho_n$. $\dot{f}/f$ is obtained from (24) using $\dot{f}/f = 3 + \frac{1}{4}\dot{\phi}^2 + \ddot{\phi}/(2\tilde{k}_n - \dot{\phi})$. This expression follows from the equations of motion, together with setting $V'/V \to \tilde{k}_n$, its value at the bounce.

## 5 Late time evolution of the interior

To completely describe the late time evolution of the interior we must patch together the sequence of bounces using the intervening Kasner epochs. We are able to do this because of the overlap between the extended bounce regime and the Kasner epochs, emphasized above and illustrated in Fig. 5. These overlaps will allow us to set up and solve a recursion relation for the

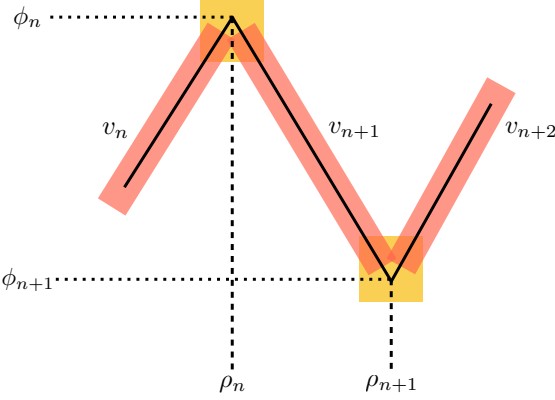

Figure 5: Overlap of the Kasner (red rectangles) and extended bounce (orange squares) regimes. The $n$th bounce is strongly localized to $\rho \approx \rho_n$ and $\phi \approx \phi_n$. However, the extended bounces described by (24) are wider and overlap with the Kasner regimes. On the $n$th Kasner regime $\dot{\phi} \approx v_n$.

locations of the bounces $\rho_n$, Kasner velocities $v_n$ and field values at the bounces $\phi_n = \phi(\rho_n)$. We shall obtain three separate relations.

Firstly, because the Kasner regimes extend to very close to the bounces we may write

$$\phi_{n+1} - \phi_n = v_{n+1}(\rho_{n+1} - \rho_n). \tag{26}$$

We will see shortly that the width of the bounce gets narrower with increasing $n$, even while the distance between successive bounces grows. Therefore, the relation (26) becomes increasingly accurate asymptotically.

Secondly, by adding the bounce solution (24) at times well before and after a bounce we obtain

$$v_n + v_{n+1} = \frac{6}{k_n} \approx 6 \frac{V'(\phi_n)}{V''(\phi_n)} \approx \frac{6}{H(\phi_n)} \,. \tag{27}$$

The final step holds to leading order at large $\phi$ due to the super-exponential potential, wherein $V'/V'' \approx V/V' = 1/H$. Equation (27) amounts to matching the extended bounce solution (24) to the Kasner regimes on either side. It may be verified to high accuracy in the numerics. However, because the full underlying equation of motion for $\phi$ is third order, we must also match the second derivative $\ddot{\phi}$ between the extended bounce and Kasner regimes. This is a little more elaborate, as we now explain.

On the Kasner regime the second derivative is exponentially small and can be obtained by differentiating (16), and setting $v_\infty \to v_n$. Does the linearized expression for $\ddot{\phi}$ obtained in this way remain valid into the extended bounce regime? As discussed around (16), the linearized solution breaks down when the potential terms become important compared to the exponential in (16). If the Kasner velocity

$$|v_n| \ll 1 \,, \tag{28}$$

then this occurs at $|H| \sim 1/|v_n|$. Because $|v_n| \ll 1$, this breakdown takes place well within the extended bounce regime $|H| \gg |v_n|$. Therefore the two descriptions overlap. Conversely, if $|v_n|$ is not small there is no parametric regime of overlap between the linearized solution and the extended bounce. Therefore, we will assume that (28) holds so that this matching of $\ddot{\phi}$ is possible. We will see that (28) indeed holds asymptotically, with $|v_n| \to 0$ as $n \to \infty$. Our analysis is therefore self-consistent in this limit. We will furthermore verify that the expressions we obtain under the assumption (28) agree with numerical results where $|v_n|$ is small but not yet tiny.

Assuming that (28) holds, from (16) we obtain, on the overlap of the Kasner and extended bounce regimes,

$$\ddot{\phi} = \tilde{a} e^{-3\phi/v_n} V'(\phi) = \hat{a} e^{-3\rho} V'(\phi) \,. \tag{29}$$

Here $\tilde{a}$ and $\hat{a}$ are constants (in a given Kasner regime). In the first equality we used the fact that $H = V'/V \gg v_n$ in the extended bounce regime to drop the $V$ term in (16). In the second equality we used the Kasner relation $\phi = v_n \rho + \text{const}$. We immediately recognize that (29) agrees perfectly with the simplified equation (20) for the extended bounce. This had to happen because we are considering a regime of overlapping validity of the two descriptions. However, the constant $c$ in (20) holds for a given extended bounce while the constant $\hat{a}$ in (29), inherited from $a$ in (16), holds through a given Kasner regime that connects two sequential bounces. It follows that the constant $c$ in (20) does not change between bounces, asymptotically, so that we may write $c = c_n = c_{n+1}$.

With the knowledge that the constant $c$ does not vary between bounces, it will be helpful to use (20) to obtain a formula for the difference in Kasner velocities across a bounce:

$$\begin{aligned}
v_{n+1} - v_n &= \int_{\rho_n^-}^{\rho_n^+} \ddot{\phi} \, d\rho = \pm c \int_{\rho_n^-}^{\rho_n^+} e^{-3\rho + \log|V'(\phi)|} d\rho \\
&\approx \pm c\sqrt{2\pi} \frac{V'(\phi_n) e^{-3\rho_n}}{\sqrt{c \, e^{-3\rho_n} V''(\phi_n)}} \,.
\end{aligned} \tag{30}$$

Here $\rho_n^\pm$ are points close to the bounce locations $\rho_n$ that are within the overlap regime. In the second step $\pm = \text{sgn}(V'(\phi))$. In the third step we have evaluated the integral using the Laplace method. The maximum of the exponent obeys $-3 + V''(\phi)/V'(\phi)\dot{\phi} = 0$. From (21), this is precisely where $\dddot{\phi} = 0$, which is at $\rho = \rho_n$. We have furthermore simplified the final

expression using the fact, cf. footnote 1, that $V'''V' \approx (V'')^2$ at large $\phi$ for super-exponential potentials. Squaring (30) we obtain

$$(v_{n+1} - v_n)^2 = \tilde{c}\, e^{-3\rho_n} \frac{V'(\phi_n)^2}{V''(\phi_n)} \approx \tilde{c}\, e^{-3\rho_n} V(\phi_n)\,. \tag{31}$$

Here $\tilde{c} = 2\pi c$ is a constant that is independent of $n$. In the final step we used that to leading order at large $\phi$ the derivative $V'' = V'^2/V$.

We may now use (26), (27) and (31) as recurrence relations to solve for the late time (large $n$) behavior of $\{\phi_n, \rho_n, v_n\}$. It is best to make the signs of the various terms explicit, as $\phi_n$ and $v_n$ are alternating in sign between epochs. From (24) it is in fact possible for the velocity not to change sign during a bounce. However, we will see that asymptotically $|k_n||\Delta v_n| \to \infty$ so that indeed each bounce leads to a change in sign, as we also saw in the numerics above. The equations to solve are then:

$$|\phi_n| + |\phi_{n+1}| = |v_{n+1}|(\rho_{n+1} - \rho_n), \quad |v_{n+1}| = |v_n| - \frac{6}{|H_n|}, \quad |v_{n+1}| + |v_n| = \sqrt{|\tilde{c}|}\, e^{-\frac{3}{2}\rho_n}\sqrt{V_n}\,. \tag{32}$$

One may eliminate $\rho_n$ and $\tilde{c}$ from these equations to obtain, in addition to the second equation in (32),

$$\log\left[\frac{|H_{n+1}|\sqrt{V_{n+1}}}{|H_n|\sqrt{V_n}} \frac{|H_n||v_n| - 3}{|H_{n+1}||v_{n+1}| - 3}\right] = \frac{3(|\phi_n| + |\phi_{n+1}|)}{2|v_{n+1}|}\,. \tag{33}$$

As $n \to \infty$ we may look for solutions that are effectively continuous functions of $n$, i.e. $v(n) = |v_n|$ and $\phi(n) = |\phi_n|$. In this limit, differences become derivatives. However, a sum may be approximated as twice the function, e.g. $|\phi_n| + |\phi_{n+1}| \approx 2\phi(n)$. In this way the recursion relations in (32) and (33) become the differential equations

$$H(\phi)\frac{dv}{dn} = -6\,, \quad \frac{d}{dn}\log\frac{H(\phi)\sqrt{V(\phi)}}{H(\phi)v - 3} = \frac{3\phi}{v}\,. \tag{34}$$

The potential $V$ is varying much more strongly than the other quantities inside the log, at large $\phi$, and therefore we may approximate the equations by

$$H(\phi)\frac{dv}{dn} = -6\,, \quad H(\phi)\frac{d\phi}{dn} = \frac{6\phi}{v}\,. \tag{35}$$

Here we used $V' = HV$.

The solution to the differential equations (35) is

$$v = \frac{b}{\phi}\,, \quad n - n_o = \frac{b}{6}\int\frac{H(\phi)}{\phi^2}d\phi\,. \tag{36}$$

Here $b$ and $n_o$ are constants of integration. It follows from (36) that if $H(\phi)$ grows at least as fast as $\phi$ as $\phi \to \infty$, then $v \to 0$ and $\phi \to \infty$ as $n \to \infty$. For example, if $H(\phi) \sim q\phi^p$ at large $\phi$, with $p > 1$, we have

$$|\phi_n| \sim \tilde{b}(n - n_o)^{1/(p-1)} \to \infty\,, \quad \text{as} \quad n \to \infty\,. \tag{37}$$

Here the constant $\tilde{b} = [6(p-1)/(bq)]^{1/(p-1)}$. When $p = 1$, $\phi \propto \log(n)$. For weakly super-exponential potentials with $0 < p < 1$, an example would be $V \sim e^{\phi^{3/2}}$, we find that $\phi$ does not diverge at late times. As many of our intermediate steps have been predicated on $\phi$ becoming large and $v$ becoming small, we will not consider these weakly super-exponential cases further here. We may note from (37) that $|k_n\Delta v_n| \approx |H_n\Delta v_n| \approx 2|H_n v_n| \propto n \to \infty$. Using the bounce solution (24), this divergence validates two statements that we made above: firstly,

that the bounces indeed become progressively narrower (from (24), $1/|k_n\Delta v_n|$ is the width of the bounce), and secondly, that indeed the Kasner velocity changes sign at each bounce (also from (24), $\dot{\phi}$ changes sign between $\rho \ll \rho_n$ and $\rho_n \ll \rho$ when $|k_n\Delta v_n|$ is large).

In Fig. 6 we verify that the asymptotic solution (36) agrees exquisitely with the numerics. We have extended the numerical range of $\rho$ plotted in §3 to capture the first 18 bounces. The asymptotic behavior (37) for $\phi_n$ matches the data very well from the second bounce on.

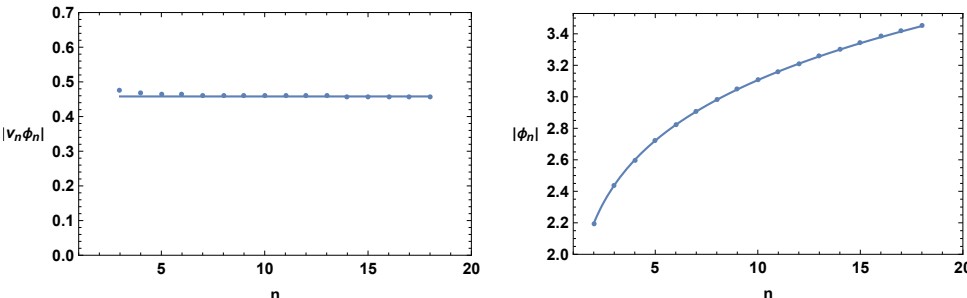

Figure 6: Behavior of $|v_n\phi_n|$ (left) and $|\phi_n|$ (right) as a function of bounce number $n$. Dots are numerical results corresponding to the potential considered previously in Fig. 1, extended to larger values of $\rho$ to capture more bounces. The solid lines are fits to the analytic expression (36). The analytic result fits both plots with only two free parameters in total: $b$ and $n_o$. The constant $b \approx 0.458$ is determined by the constant value of $|v_n\phi_n|$. The field values $|\phi_n|$ are then fit to (37) with $p = 7$ and $\tilde{b} \approx 2.15$ fixed, leading to $n_o \approx 0.86$.

The above arguments have proven that, with sufficiently strong super-exponential potentials, the Kasner velocity relaxes all the way to zero and hence the interior tends towards the Schwarzschild singularity through an infinite sequence of Kasner epochs. We may also ask how long the Kasner epochs are. From (32) we have

$$\frac{d\rho}{dn} = \frac{2\phi}{v} \quad \implies \quad \rho_n \sim \hat{b}\,(n-n_o)^{(p+1)/(p-1)}, \tag{38}$$

with $\hat{b}$ a constant. This expression is again in excellent agreement with the numerical data.

A final technical comment is in order, to further shore up confidence in the results above. It may be verified on the numerical solutions that the final recurrence relation in (32) is not precisely obeyed, with $\tilde{c} = 2\pi c$ varying slightly from bounce to bounce. This variation is due to the $e^{-\frac{1}{4}v_n\phi}$ term in the linearized solution (16) for the Kasner regime. We had dropped that term previously in (29) because it was subleading compared to the $e^{-3\phi/v_n}$ term at $|v_n| \ll 1$. We may re-instate this term and re-run the matching argument between the Kasner and extended bounce regimes. To capture the leading order correction to the results above it is sufficient, in matching the solutions, simply to evaluate $e^{-\frac{1}{4}v_n\phi}$ at the endpoints of the Kasner regime $\phi \approx \phi_{n-1}$ and $\phi \approx \phi_n$, respectively. This leads to the variation

$$\log \frac{c_{n+1}}{c_n} = -\frac{1}{4}(\rho_{n+1} - \rho_n)v_{n+1}^2. \tag{39}$$

We have checked that this expression for $c_{n+1}/c_n$ agrees very well with the numerically seen variation of $c$ between bounces.

The variation of the constant between bounces corrects the recurrence relations above by adding one half times (39) to the left hand side of (33). In the large $n$ limit (39) becomes $\frac{1}{4}(\partial_n\rho)v^2 = \frac{1}{2}\phi v$, using (38). Asymptotically this is much smaller than the right hand side of (33), which is $3\phi/v$. Therefore, this extra term is negligible and the variation of the constant $c$ between bounces does not alter the late time asymptotic behavior of the interior.

## 6 Discussion

We have explained that a scalar field with a super-exponential potential is a simple way to realize an infinite sequence of Kasner epochs in the interior of an asymptotically AdS black hole. We have been able to characterize the late interior time behavior rather explicitly. Our main objective has been to develop a setting that may be useful for exploring the holographic meaning of interior dynamics. However, it may also be interesting to investigate the approach to singularities with super-exponential potentials more generally. The most fully characterized 'cosmological billiards' [7, 30], involves bounces off weaker, exponential barriers only. Those exponential barriers — or 'walls' — originate from gravitational kinetic energy and curvature terms as well as explicit potentials for matter fields.

Many dimensional reductions of string theory lead to exponential potentials for the various Kaluza-Klein scalar fields. It may be interesting to understand how super-exponential potentials can be realized in microscopic string-theoretic settings. This may allow interesting stringy effects to arise towards the singularity in a controlled way, as the scalar field probes higher and higher up its potential with each bounce. Relatedly, it may be interesting to understand how various 'swampland' conjectures such as [34, 35], that constrain the behavior of scalar potentials, relate to the approach to spacelike singularities.

In the solutions that we have discussed the scalar field bounces back and forth in its potential, but the metric component $g_{tt}$ relaxes monotonically towards the Schwarzschild singularity. This can be contrasted with the 'Kasner inversions' described in [14] where $g_{tt}$ itself experiences alternating period of growth and collapse. It would be interesting to find a simple model, possibly along the lines of [16, 28, 29], exhibiting an infinite number of bounces of $g_{tt}$ towards the singularity.

## Acknowledgements

The work of SAH is partially supported by Simons Investigator award #620869 and by STFC consolidated grant ST/T000694/1. NN was supported through the CMS Summer Research Bursary Fund.

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
