# Peer review of "AdS Black Holes with a Bouncing Interior"

_SciPost Physics, doi:SciPost Phys. 14, 074 (2023)_

## Round 2 · Referee Report · Anonymous (Referee 1) · 2022-10-27

Report

The authors construct a concrete, solvable model of classical interior black hole dynamics. Their model builds on a long history of work in this area. While it has been known for a long time that some classical black hole interiors with exponential potentials consist of an infinite series of Kasner epochs as we flow towards the singularity, the authors construct a simple class of models in which this type of evolution can be analyzed analytically for the case of superexponential potentials. This work is certainly novel and useful.

However, I have some points which I would like the authors to address before I recommend publication. I have also included any observed typos in my report.

Requested changes

  1. Below equation (3), note that this potential (or, being more specific, setting $m^2 = -2$) does not necessarily fix $\Delta = 2$. One may also consider the quantization with $\Delta = 1$ for 3 boundary dimensions, which corresponds to a different boundary condition for the scalar (cf. Klebanov-Witten arXiv:9905104). This ultimately does not adversely impact the analysis since the authors set the boundary condition in (4), but they should be clear that they are actually choosing the $\Delta = 2$ quantization explicitly.

  2. Equation (16) is unable to pass my own consistency check. Specifically, I start with equation (12), switching from $H$ ot $V$ by substituting $H \to V'/V$, $H' \to (V/V')'$ (inverting (8)) and plugging-in the ansatz $v(\phi) = v_\infty + \delta v(\phi)$. I then expand around $\delta v(\phi) \approx 0$ and truncate at leading order, thereby leaving me with a differential equation relating $\delta v'(\phi)$ and $\delta v''(\phi)$. I would expect (16) to solve the resulting equation, but I find that this is not the case. This means that either I am mistaken somewhere in my approach or (16) is not correct as written. As such, I would appreciate it if the authors could clarify their derivation of (16).

  3. I definitely agree that Figure 3 is sufficient to show that there is both a narrow bounce in $V$ and a more extended bounce in $H$. However, I do not think that the current presentation is the simplest way to demonstrate the characterization of the extended bounce regime as that for which $|H| \gg |v|$, since this plot only covers a small $\Delta\rho = \pm 1$ window around the bounce. I think the simplest way to satisfy my complaint here would be to zoom Figure 3 out to a larger domain for $\rho - \rho_H$ which encapsulates more of the pre- and post-bounce behavior. Furthermore, so that the point about $V$ having a very narrow spike is not lost, I believe the authors could do something like in Figure 2 where they embed the plots of $H$ and $V$ over a narrower domain within a plot of the same functions over a wider domain.

  4. In the paragraph below (38), the authors arrive at the bounces becoming narrower and the Kasner velocity changing velocity at each bounce in the late-time regime, with the justification being that $|k_n| |\Delta v_n| \to \infty$. However, why this is enough is still unclear to me. I would have thought that the signature of the former phenomenon would be a shortening in the width of the extended region, whereas the signature of the latter would be that $\Delta v_n$ changes sign for each $n$. How can I read off this behavior from the stated limiting behavior of $|k_n| |\Delta v_n|$?

TYPOS: - pg. 3: Above (1), ...the Lagrangian density" $\to$...the Lagrangian density is" - pg. 12: Above (32), the authors write $V''' V' \approx (V')^2$ for large $\phi$ and superexponential potentials. This should read $V''' V' \approx (V'')^2$.

---

## Round 2 · Referee Report · Anonymous (Referee 2) · 2022-11-2

Report

In this paper, the authors find analytic solutions for spatially homogeneous black hole interiors in AdS gravity coupled to a scalar field. The scalar field oscillates in its potential, producing an infinite sequence of bounces between Kasner-like solutions.

This is an interesting setup because it provides an analytic playground for the study of classical dynamics near a singularity, in a setting that could plausibly have a holographic description. It is well written and thoroughly checked with numerics vs analytics. I recommend this article for publication.

---

## Round 3 · Referee Report · Anonymous · 2023-1-3

Report

The authors have appropriately addressed all of my concerns. I am now happy to recommend this paper for publication.

---

## Round 3 · Author Response

We thank both of the referees for taking the time to look at the paper and their positive comments.

Referee 2 has recommended publication in its current form.

Referee 1 has raised some minor points which we have addressed as described below.

We have also corrected the two typos found by Referee 1.

---

## Round 3 · List of Changes

1. The referee correctly recalled there are two possible quantizations of the scalar field when m2=-2. We have therefore added “, using standard quantization” to the sentence below equation (3). As the referee has noted the falloff is given explicitly in equation (4).

2. We have pasted below Mathematica code that shows that (16) is correct as written in the paper. The steps are literally those given in the paper and outlined by the referee, so we do not believe that a clarification in the text is necessary. Most likely the referee has made a typo writing down the equation.

(* this is equation (12) *)

In[1]:= eq=4(2 H[\[Phi]]-v[\[Phi]])v[\[Phi]]v''[\[Phi]]/v'[\[Phi]]== (8 H[\[Phi]]^2 v[\[Phi]]- 2H[\[Phi]](3 v[\[Phi]]^2+4(3+v'[\[Phi]]))+v[\[Phi]](12 + v[\[Phi]]^2 + 4D[2 H[\[Phi]]-v[\[Phi]],\[Phi]]));

In[2]:= vv[\[Phi]_] = vo + \[Delta] dv[\[Phi]];

(* this is the linearized equation *)

In[3]:= eq1=Series[eq /. v -> vv,{\[Delta],0,0}] // Normal// FullSimplify

Out[3]= 6 (4+vo^2) H[\[Phi]]==vo (12+vo^2+8 H[\[Phi]]^2+8 (H^\[Prime])[\[Phi]]+(4 (vo-2 H[\[Phi]]) (dv^\[Prime]\[Prime])[\[Phi]])/(dv^\[Prime])[\[Phi]])

(* this is the expression written in (16), directly for the derivative*)

In[4]:= dvp[\[Phi]_] = a Exp[- (3 + vo^2/4)\[Phi]/vo](V[\[Phi]]- 2/vo V'[\[Phi]]);

(* here we verify that the expression (16) solves the linearized equation *)

In[5]:= eq1 /. dv'[\[Phi]] -> dvp[\[Phi]] /. dv''[\[Phi]] -> dvp'[\[Phi]] /. V''[\[Phi]] -> D[H[\[Phi]] V[\[Phi]],\[Phi]] /. V'[\[Phi]] -> H[\[Phi]] V[\[Phi]]// FullSimplify

Out[5]= True

3. The referee is concerned that Fig 3 is insufficient to demonstrate that |H| >> |v| over the extended bounce regime because it only covers a narrow window of rho. However, if we look at figure 4, which is shown for the same bounce, we see that Kasner behavior is recovered on either side over an even narrower window of rho. Therefore Fig 3 and Fig 4 combined indeed show that |H| >> |v| holds over a regime that extends from within one Kasner regime to within the next. The overlap with the Kasner regimes is the crucial point. There is not much to be gained from extending the plot range of Fig 3 as suggested by the referee. However, we do understand the referee’s concern and have therefore added “, see Fig. 4 below,” in the sentence underneath equation (17).

4. As we had already stated in the text, both statements that we made at end of the paragraph below (38) follow from equation (24). However, we do agree that the explanation here was a little compressed. We have added the clarifying comment “(from (24), 1/|k_n Delta v_n| is the width of the bounce)” and “(also from (24), phi-dot changes sign between rho >> rho_n and rho << rho_n when |k_n Delta v_n| is large)”.

---

## Editorial Decision

published